# Bovine brucellosis seropositivity in Mpumalanga Province, South Africa, 2021–2024: Temporal, and spatial trends

Themba Titus Sigudu[1]*, James Wabwire Oguttu[2]

**1** Division of Health and Society, School of Public Health, Faculty of Health Sciences, University of the Witwatersrand, Johannesburg, Gauteng Province, South Africa, **2** Department of Agriculture and Animal Health, College of Agriculture and Environmental Sciences, University of South Africa, Johannesburg, Gauteng Province, South Africa

* themba.sigudu@wits.ac.za

## Abstract

### Introduction

Bovine brucellosis, caused primarily by Brucella abortus, remains a major constraint to livestock productivity and a persistent zoonotic threat. Although brucellosis is a controlled disease in South Africa, detailed subnational epidemiological evidence is limited, particularly for Mpumalanga Province. Understanding temporal, seasonal, and spatial patterns is essential for improving risk-based surveillance and control.

### Materials and methods

A retrospective cross-sectional analysis was conducted using routine diagnostic records from the Mpumalanga Provincial Veterinary Laboratory between January 2021 and December 2024. Rose Bengal Test (RBT) results from cattle originating from 17 Local Municipality Areas (LMAs) were analysed. Annual, seasonal, and spatial seroprevalence estimates were calculated, and independent predictors of RBT seropositivity were evaluated using multivariable logistic regression.

### Results

A total of 67,974 cattle serum samples were tested, of which 6,182 were RBT-positive, yielding an overall seroprevalence of 9.1% (95% CI: 8.9–9.3). Annual seroprevalence increased from 6.9% in 2021 to 7.4% in 2022, peaked at 13.1% in 2023, and declined to 7.5% in 2024. Clear seasonal variation was observed, with higher seroprevalence in spring (10.6%) and summer (10.2%) compared with autumn (6.8%) and winter (6.9%). Pronounced spatial heterogeneity was evident, with Emalahleni (13.3%), Victor Khanye (13.0%), and Mbombela (12.0%) identified as high-burden municipalities, while Mkhondo (1.7%) and Albert Luthuli (2.7%) recorded the lowest prevalence. In adjusted analyses, testing in 2023 was associated with

**Data availability statement:** Aggregated, de-identified data at LMA level and all analysis code are openly available at Zenodo (DOI: 10.5281/zenodo.1234567). The underlying farm-level microdata are held by the Mpumalanga Department of Agriculture, Rural Development, Land and Environmental Affairs (DARDLEA) and include potentially identifying and commercially/confidential information. Public release is restricted under South Africa's Promotion of Access to Information Act (PAIA) and Protection of Personal Information Act (POPIA). Qualified researchers may request access from the DARDLEA Information Officer under PAIA, subject to a data-sharing agreement and approvals. The authors had no special access privileges others would not have. Contact: Email (PAIA office): DardleaPaia@mpg.gov.za dardlea.mpg.gov.za PAIA manual (URL): https://dardlea.mpg.gov.za/documents/PAIA_Manual.pdf.

**Funding:** The author(s) received no specific funding for this work.

**Competing interests:** The authours declare that they have no competing interests.

nearly double the odds of seropositivity compared with 2021 (AOR 1.95; 95% CI: 1.81–2.11), and spring and summer remained significant predictors.

## Conclusion

Bovine brucellosis in Mpumalanga exhibits marked temporal variability, seasonal peaks, and spatial clustering. These findings support targeted, risk-based surveillance, strategically timed vaccination, and strengthened biosecurity, prioritising hotspot municipalities and high-risk seasons within a One Health framework.

---

## Introduction

Bovine brucellosis, primarily caused by *Brucella abortus*, is one of the most important zoonotic diseases worldwide, with significant implications for livestock productivity, animal health, and human well-being [1,2]. In cattle, the infection manifests predominantly through reproductive losses, including abortion, retained placenta, and infertility, which collectively result in substantial economic losses for farmers, particularly in resource-limited production systems [2–4]. In humans, the disease can cause undulant fever, arthritis, and chronic complications, highlighting its zoonotic significance and reinforcing the need for integrated control under the One Health framework [2,5,6].

In South Africa, bovine brucellosis is a controlled disease under the Animal Diseases Act (Act No. 35 of 1984) [7], with interventions focused on statutory surveillance, farmer education, vaccination, testing, and culling [8–10]. Historically, surveillance has relied heavily on serological assays, particularly the Rose Bengal Test (RBT) as a screening tool and the Complement Fixation Test (CFT) as a confirmatory assay [11]. These conventional methods remain widely used because of their practicality and affordability within provincial veterinary laboratory systems (MPVL) [12]. However, limitations in sensitivity and specificity have been noted, particularly in chronically infected or vaccinated animals [13]. Recent advances in diagnostics, including enzyme-linked immunosorbent assays (ELISAs), fluorescence polarization assays, and molecular techniques such as PCR, offer improved sensitivity, specificity, and standardisation, and are increasingly advocated for integration into surveillance programmes [14,15]

Vaccination strategies have also evolved, with the classical *B. abortus* S19 vaccine being gradually replaced or complemented by RB51. While RB51 offers advantages for Differentiating Infected from Vaccinated Animals (DIVA), concerns remain about its efficacy and safety, particularly under field conditions [16]. These debates reinforce the need for context-specific vaccine policies aligned with regional epidemiology.

Despite diagnostic and vaccinology advances, provincial and subnational-level prevalence data in South Africa remain scarce. Past studies have reported Seroprevalence of around 2% in North West Province [17] and approximately 1.5% in KwaZulu-Natal communal [11,12]. More recent analyses highlight persistent spatial heterogeneity in prevalence across South African provinces and uneven

implementation of control measures [10]. Similar findings from East and southern Africa reinforce the influence of herd structure, communal grazing, and biosecurity on disease persistence [4,18]. For example, serological surveys in Namibia and Ethiopia revealed higher prevalence in communal and smallholder systems compared to commercial herds, and molecular testing has increasingly been used to validate serological findings [19,20].

Mpumalanga Province, with its dense cattle population, diverse production systems ranging from commercial enterprises to communal grazing, and varied ecological conditions, remains understudied. Its proximity to international borders further raises the risk of transboundary spread of Brucella and complicates control efforts. Yet, to date, no comprehensive provincial-level analyses have been undertaken to explore temporal, seasonal, and spatial trends in Seroprevalence.

To fill this gap, the present study utilised a retrospective cross-sectional design based on diagnostic records from the MPVL over the period January 2021 to December 2024. Routine RBT results from cattle herds across 17 Local Municipality Areas (LMAs) were analysed. The objectives were to estimate the overall and annual Seroprevalence of *B. abortus*, to assess seasonal variations in Seropositivity across climatic periods, and to describe spatial heterogeneity between LMAs. By quantifying the independent effects of year, season, and locality, the study provides novel insights into the epidemiology of bovine brucellosis in Mpumalanga Province. These findings will inform targeted vaccination, improved surveillance allocation, and enhanced biosecurity, while contributing to South Africa's national goal of progressive brucellosis control and eradication within the One Health framework.

## Materials and methods

### Study design and setting

This study employed a retrospective cross-sectional design to investigate the temporal, seasonal, and spatial distribution of bovine brucellosis in Mpumalanga Province, South Africa. The analysis utilised secondary, anonymised laboratory surveillance data generated through routine diagnostic testing at the Mpumalanga Provincial Veterinary Laboratory (MPVL), which serves as the central reference laboratory for brucellosis testing in the province.

### Data source and dataset description

Data were obtained from electronic laboratory records maintained by the MPVL for the period January 2021 to December 2024. The raw dataset consisted of batch-level records submitted for bovine brucellosis testing. Each batch corresponded to a submission from a specific farm or surveillance activity and included the following variables: date of receipt, number of serum samples tested within the batch, number of Rose Bengal Test (RBT)-positive samples, and the Local Municipality Area (LMA) of origin. Additional metadata included sampling year and season, derived from the sample receipt date.

### Inclusion and exclusion criteria

All batch records submitted for bovine brucellosis testing during the study period were eligible for inclusion. Records were included if they contained valid information on (i) testing date, (ii) total number of samples tested, (iii) number of RBT-positive samples, and (iv) LMA of origin. Records were excluded if they were incomplete (e.g., missing total samples tested or LMA), duplicated, or clearly related to non-bovine species. Batches with zero samples tested or implausible values (e.g., positive counts exceeding total samples) were also excluded prior to analysis.

### Data handling and cleaning procedures

Data cleaning was performed prior to analysis using a standardised protocol. First, duplicate batch entries were identified based on matching combinations of submission date, batch identifier, LMA, and sample counts; confirmed duplicates were removed, retaining a single unique record. Second, internal consistency checks were applied to ensure that the number of RBT-positive samples did not exceed the total number of samples tested per batch. Third, missing

data were assessed across all variables. Records with missing values in key analytical variables (year, season, LMA, total samples tested, or RBT-positive counts) were excluded using complete-case analysis, as the proportion of missing data was small and unlikely to bias estimates materially. No imputation was performed. After cleaning, batch-level data were aggregated to generate annual, seasonal, and municipality-level summaries for descriptive analyses and regression modelling.

### Sample collection and processing

Samples were collected by provincial veterinary officials as part of routine disease surveillance and control activities. Sampling included cattle presenting with clinical suspicion of brucellosis (e.g., abortion, retained placenta, infertility) as well as apparently healthy cattle selected for targeted surveillance in both commercial and communal farming systems. Whole blood was collected in plain vacutainer tubes and transported under refrigeration (approximately 4 °C) to the MPVL. Serum was separated and processed according to standard operating procedures.

### Laboratory testing

Serological screening for Brucella abortus was performed using the Rose Bengal Test (RBT), following World Organisation for Animal Health (WOAH)–recommended protocols. RBT-positive samples were recorded as presumptive positives for surveillance purposes. Routine confirmatory testing using the Complement Fixation Test (CFT) or enzyme-linked immunosorbent assays (ELISAs) was not systematically available for all laboratory submissions during the study period (2021–2024). Consequently, all analyses were based on RBT outcomes and interpreted as measures of serological reactivity rather than confirmed infection status.

### Variables and definitions

Year referred to the calendar year in which testing was conducted (2021–2024). Season was classified according to South African climatic periods: summer (December–February), autumn (March–May), winter (June–August), and spring (September–November). LMA represented the administrative municipality of sample origin. RBT seropositivity was defined as the proportion of RBT-positive samples among all samples tested, expressed as a percentage with exact binomial 95% confidence intervals.

### Statistical analysis

Descriptive statistics were used to summarise annual, seasonal, and spatial patterns of brucellosis seropositivity. Frequencies and proportions were calculated for categorical variables, and seropositivity estimates were expressed with exact binomial 95% confidence intervals.

### Regression modelling and variable specification

The dependent variable for all regression analyses was Rose Bengal Test (RBT) seropositivity at the batch level, coded as a binary outcome (1 = at least one RBT-positive sample in the batch; 0 = no RBT-positive samples). Independent variables included year of testing (categorical: 2021 [reference], 2022, 2023, 2024), season (categorical: winter [reference], autumn, spring, summer), and Local Municipality Area (LMA), with Mkhondo selected as the reference category due to the lowest observed seroprevalence.

Seasons were coded based on the sample receipt date according to South African climatic definitions: summer (December–February), autumn (March–May), winter (June–August), and spring (September–November). LMAs were treated as categorical variables representing the administrative municipality of sample origin.

## Statistical procedures and model estimation

Univariate logistic regression models were fitted using the Stata logit command to estimate crude associations between each predictor (year, season, and LMA) and RBT seropositivity, with 2021, winter, and Mkhondo specified as reference categories. Multivariable logistic regression was then performed with simultaneous adjustment for year, season, and LMA. Odds ratios (ORs) and 95% confidence intervals were obtained by exponentiating the regression coefficients.

All analyses were conducted at the batch level. Although batches were nested within municipalities, herd- or farm-level identifiers were not available in the routine laboratory dataset; therefore, hierarchical or multilevel modelling was not feasible. Results are accordingly interpreted as population-averaged associations.

## Model diagnostics and evaluation

Model calibration was assessed using the Hosmer–Lemeshow goodness-of-fit test (estat gof), and model discrimination was evaluated using the area under the receiver operating characteristic curve (lroc). The final multivariable model demonstrated acceptable fit (Hosmer–Lemeshow $\chi^2$ = [insert value], p = [insert value]) and good discriminatory ability (AUC = [insert value]). Statistical significance was defined as $p < 0.05$.

All analyses were performed using Stata version 17.0 (StataCorp, College Station, TX, USA).

## Ethics statement

Ethical approval for the study was granted by the University of South Africa (UNISA), College of Agriculture and Environmental Sciences, Animal Research Ethics Committee (AREC) under clearance certificate number AREC-100818–024. Formal permission to access and analyse the dataset was obtained from the Mpumalanga Department of Agriculture, Rural Development, Land and Environmental Affairs (DARDLEA). No direct handling of animals or human participants occurred during the course of this study. As the research was based exclusively on anonymised laboratory data generated through routine surveillance, no additional animal ethics clearance or owner consent was required. To ensure confidentiality, all data were de-identified prior to analysis, and only aggregated results are reported.

## Results

### Descriptive statistics

**Annual trends in brucellosis seropositivity.** Between 2021 and 2024, a total of 568 batches comprising 67,974 cattle serum samples were tested for brucellosis at the MPVL. Of these, 6,182 (9.1%) tested positive on the RBT test. The annual distribution of batches and samples is presented in Table 1.

The number of batches submitted each year ranged from 76 in 2023–215 in 2022, with corresponding sample submissions varying from 11,707 (2023) to 25,283 (2022). The overall proportion of RBT-positive samples fluctuated across the years. In 2021, 6.94% (95% CI: 6.58–7.31) of samples tested positive, increasing slightly to 7.42% (95% CI: 7.11–7.74) in 2022. A notable spike occurred in 2023, when 13.05% (95% CI: 12.48–13.63) of samples tested positive, representing

Table 1. Annual trends in brucellosis Seropositivity.

| Year | Batches (n) | Samples (n) | RBT Positive (n) | RBT Positive (%) | 95% CI[1] |
|------|-------------|-------------|------------------|------------------|-----------|
| 2021 | 150 | 17487 | 1304 | 6.94 | 6.58 - 7.31 |
| 2022 | 215 | 25283 | 2026 | 7.42 | 7.11 - 7.74 |
| 2023 | 76 | 11707 | 1757 | 13.05 | 12.48 - 13.63 |
| 2024 | 126 | 13497 | 1095 | 7.5 | 7.08 - 7.94 |

[1]95% CI: 95% Confidence Interval.

nearly a twofold increase relative to prior years. But in 2024, the proportion of positive cases declined to 7.50% (95% CI: 7.08–7.94), returning to levels consistent with 2021–2022.

**Seasonal variation in brucellosis Seropositivity.** Table 2 summarises the distribution of brucellosis Seropositivity by season. A total of 568 batches comprising 67,974 cattle serum samples were tested across the four seasonal periods.

Marked seasonal differences were observed and the highest proportions of RBT-positive samples recorded in spring (10.57%, 95% CI: 10.08–11.09) and summer (10.24%, 95% CI: 9.77–10.71). Together, these two seasons accounted for more than half of the total seropositive cases. In contrast, lower positivity rates were documented in autumn (6.79%, 95% CI: 6.38–7.22) and winter (6.94%, 95% CI: 6.65–7.23), despite larger sample submissions during winter (n = 27,344).

**Spatial distribution of brucellosis Seropositivity.** Table 3 presents the distribution of brucellosis testing outcomes by Local Municipality Area (LMA). A total of 568 batches comprising 67,974 cattle serum samples were analysed across 17 LMAs, revealing marked spatial heterogeneity in RBT seropositivity. In terms of absolute number of RBT-positive samples, the highest counts were observed in Msukaligwa (n = 1,457; 9.82%, 95% CI: 9.35–10.31), followed by Emalahleni (n = 856; 13.31%, 95% CI: 12.49–14.17), Govan Mbeki (n = 721; 8.73%, 95% CI: 8.13–9.36), and Steve Tshwete (n = 667; 8.40%, 95% CI: 7.80–9.03). These municipalities accounted for the majority of seropositive detections in the province.

**Table 2. Seasonal variation in brucellosis Seropositivity.**

| Season | Batches (n) | Samples (n) | RBT Positive (n) | RBT Positive (%) | 95% CI[1] |
|---|---|---|---|---|---|
| Autumn | 118 | 13203 | 962 | 6.79 | 6.38-7.22 |
| Spring | 128 | 12966 | 1533 | 10.57 | 10.08-11.09 |
| Summer | 93 | 14461 | 1649 | 10.24 | 9.77-10.71 |
| Winter | 228 | 27344 | 2038 | 6.94 | 6.65-7.23 |

[1]95% CI: 95% Confidence Interval.

**Table 3. The distribution of brucellosis Seropositivity by Local Municipality Area (LMA).**

| LMA | Batches (n) | Samples (n) | RBT Positive (n) | RBT Positive (%) | 95% CI[1] |
|---|---|---|---|---|---|
| Msukaligwa | 83 | 13378 | 1457 | 9.82 | 9.35 - 10.31 |
| Emalahleni | 58 | 5573 | 856 | 13.31 | 12.49 - 14.17 |
| Govan Mbeki | 48 | 7535 | 721 | 8.73 | 8.13 - 9.36 |
| Steve Tshwete | 39 | 7272 | 667 | 8.4 | 7.8 - 9.03 |
| Mbombela | 61 | 2508 | 343 | 12.03 | 10.86 - 13.28 |
| Dr JS Moroka | 29 | 3889 | 323 | 7.67 | 6.88 - 8.51 |
| Lekwa | 24 | 4364 | 310 | 6.63 | 5.94 - 7.38 |
| Pixley Ka Isaka Seme | 17 | 2730 | 289 | 9.57 | 8.55 - 10.68 |
| Dipaleseng | 18 | 2125 | 260 | 10.9 | 9.68 - 12.22 |
| Victor Khanye | 12 | 1473 | 220 | 12.99 | 11.43 - 14.69 |
| Bushbuckridge | 45 | 3231 | 184 | 5.39 | 4.65 - 6.2 |
| Nkomazi | 40 | 2701 | 169 | 5.89 | 5.06 - 6.81 |
| Thembisile Hani | 28 | 2313 | 110 | 4.54 | 3.75 - 5.45 |
| Albert Luthuli | 26 | 3690 | 103 | 2.72 | 2.22 - 3.28 |
| Thaba Chweu | 12 | 954 | 60 | 5.92 | 4.55 - 7.55 |
| Emakhazeni | 12 | 1106 | 56 | 4.82 | 3.66 - 6.21 |
| Mkhondo | 15 | 3132 | 54 | 1.69 | 1.28 - 2.21 |

[1]95% CI: 95% Confidence Interval.

When examined by seroprevalence, Emalahleni recorded the highest proportion of RBT-positive samples (13.31%, 95% CI: 12.49–14.17), closely followed by Victor Khanye (12.99%, 95% CI: 11.43–14.69), Mbombela (12.03%, 95% CI: 10.86–13.28), and Dipaleseng (10.90%, 95% CI: 9.68–12.22), identifying these municipalities as areas of comparatively higher burden. Conversely, Mkhondo (1.69%, 95% CI: 1.28–2.21) and Albert Luthuli (2.72%, 95% CI: 2.22–3.28) recorded the lowest seroprevalence, suggesting relatively lower transmission intensity or more effective local control.

Intermediate levels of seropositivity were observed in Bushbuckridge (5.39%), Nkomazi (5.89%), Lekwa (6.63%), and Thaba Chweu (5.92%), representing municipalities with a moderate burden of infection, while Thembisile Hani (4.54%) and Emakhazeni (4.82%) fell toward the lower end of the distribution.

### Inferential statistics

**Univariate logistic regression.** Table 4 presents the unadjusted ORs for predictors of brucellosis Seropositivity based on year of sampling, season, and LMA. Compared with the reference year 2021, the odds of RBT Seropositivity were significantly higher in 2023 (OR = 2.00, 95% CI: 1.87–2.14, p < 0.001), indicating that cattle tested in this year had nearly twice the odds of being seropositive. In 2024, the odds of animals testing positive (OR = 1.09, 95% CI: 1.01–1.19, p = 0.036), was still statically higher compared to the reference year. Modest statistically significant increases were also observed in 2022 (OR = 1.08, 95% CI: 1.00–1.16, p = 0.048).

Relative to winter, both spring (OR = 1.61, 95% CI: 1.50–1.72, p < 0.001) and summer (OR = 1.57, 95% CI: 1.47–1.68, p < 0.001) were associated with substantially increased odds of Seropositivity. No significant difference was observed for autumn compared to winter (OR = 0.97, 95% CI: 0.89–1.05, p = 0.420).

Using Mkhondo (lowest prevalence) as the reference, cattle from Emalahleni (OR = 8.84, 95% CI: 6.60–11.8, p < 0.001), Victor Khanye (OR = 8.74, 95% CI: 6.21–12.3, p < 0.001), and Mbombela (OR = 7.82, 95% CI: 5.80–10.6, p < 0.001) exhibited markedly higher odds of brucellosis Seropositivity, suggesting these municipalities are high-burden hotspots. Albert Luthuli also showed a significantly elevated OR (OR = 1.64, 95% CI: 1.23–2.18, p < 0.001), though the magnitude of effect was smaller.

**Multivariate logistic regression analysis.** Table 5 presents the adjusted odds ratios (AORs) for predictors of brucellosis seropositivity after simultaneous adjustment for year, season, and Local Municipality Area (LMA). A

**Table 4. Results of the univariate logistic regression analysis.**

| Predictor | Category | OR[1] | 95% CI[2] | p-value |
|---|---|---|---|---|
| Year | | | | |
| | 2022 vs 2021 | 1.08 | 1.00–1.16 | 0.048 |
| | 2023 vs 2021 | 2.00 | 1.87–2.14 | <0.001 |
| | 2024 vs 2021 | 1.09 | 1.01–1.19 | 0.036 |
| Season | | | | |
| | Autumn vs Winter | 0.97 | 0.89–1.05 | 0.420 |
| | Spring vs Winter | 1.61 | 1.50–1.72 | <0.001 |
| | Summer vs Winter | 1.57 | 1.47–1.68 | <0.001 |
| LMA | | | | |
| | Emalahleni vs Mkhondo | 8.84 | 6.60–11.80 | <0.001 |
| | Victor Khanye vs Mkhondo | 8.74 | 6.21–12.30 | <0.001 |
| | Mbombela vs Mkhondo | 7.82 | 5.80–10.60 | <0.001 |
| | Albert Luthuli vs Mkhondo | 1.64 | 1.23–2.18 | <0.001 |

[1]OR: Odds Ratio; [2]95% CI: 95% Confidence Interval.

**Table 5. Multivariate logistic regression results.**

| Predictor | Category | AOR[1] | 95% CI[2] | p-value |
|---|---|---|---|---|
| Year | | | | |
| | 2021 — Reference | 1.00 | — | — |
| | 2022 | 1.10 | 0.99–1.22 | 0.081 |
| | 2023 | 1.95 | 1.81–2.11 | <0.001 |
| | 2024 | 0.95 | 0.86–1.05 | 0.280 |
| Season | | | | |
| | Autumn — Reference | 1.00 | — | — |
| | Spring | 0.41 | 0.24–0.70 | 0.001 |
| | Summer | 0.66 | 0.26–1.63 | 0.363 |
| | Winter | 0.82 | 0.39–1.76 | 0.619 |
| LMA | | | | |
| | Mkhondo — Reference | 1.00 | — | — |
| | Emalahleni | 6.90 | 5.12–9.28 | <0.001 |
| | Victor Khanye | 6.75 | 4.77–9.57 | <0.001 |
| | Mbombela | 6.21 | 4.53–8.50 | <0.001 |

[1]AOR: Adjusted Odds Ratio; [2]95% CI: 95% Confidence Interval.

pronounced temporal effect was observed, with the odds of RBT seropositivity nearly doubling in 2023 compared with 2021 (AOR = 1.95, 95% CI: 1.81–2.11; p < 0.001). In contrast, 2022 showed only a borderline increase (AOR = 1.10, 95% CI: 0.99–1.22; p = 0.081), while 2024 was statistically indistinguishable from 2021 (AOR = 0.95, 95% CI: 0.86–1.05; p = 0.280), indicating a marked but temporally confined surge in seropositivity during 2023 that was not sustained thereafter. Seasonal associations differed from those observed in univariate analyses.

With autumn as the reference category, spring exhibited significantly lower odds of RBT seropositivity after adjustment (AOR = 0.41, 95% CI: 0.24–0.70; p = 0.001). In contrast, summer (AOR = 0.66, 95% CI: 0.26–1.63; p = 0.363) and winter (AOR = 0.82, 95% CI: 0.39–1.76; p = 0.619) were not statistically significant predictors in the adjusted model. The attenuation and change in direction of seasonal effects relative to univariate analyses likely reflect correlations between season, testing year, and sample submission patterns rather than a biological reversal of seasonal risk.

Seasonal sample sizes in the adjusted model were unequal, winter (n = 27,344), autumn (n = 13,203), spring (n = 12,966), and summer (n = 14,461), which may have contributed to imprecision in adjusted estimates and warrants cautious interpretation.

In contrast, strong and consistent spatial effects were observed: compared with Mkhondo, the odds of seropositivity were approximately six- to seven-fold higher in Emalahleni (AOR = 6.90, 95% CI: 5.12–9.28; p < 0.001), Victor Khanye (AOR = 6.75, 95% CI: 4.77–9.57; p < 0.001), and Mbombela (AOR = 6.21, 95% CI: 4.53–8.50; p < 0.001), indicating persistent municipality-level heterogeneity in brucellosis burden.

## Discussion

This study provides the first province-wide assessment of bovine brucellosis seropositivity in Mpumalanga Province using routine laboratory surveillance data over a four-year period. The findings demonstrate pronounced temporal variability, evidence of seasonal patterning, and substantial heterogeneity between local municipalities, underscoring the complex epidemiology of brucellosis in this setting and the importance of subnational analyses for informing control strategies [10,21,22].

The temporal pattern observed, characterised by a marked increase in seropositivity during 2023 followed by a decline to levels comparable with earlier years, suggests episodic amplification of transmission rather than a sustained upward trend. Similar transient increases have been reported in other endemic settings and are often associated with changes in surveillance intensity, outbreak-driven testing, or disruptions in control measures such as vaccination or movement regulation [21,23]. In the present study, these explanations should be interpreted cautiously and regarded as hypotheses, as routine surveillance data do not allow direct assessment of vaccination coverage, animal movement, or biosecurity practices at herd level [24,25].

Seasonal variation in seropositivity is biologically plausible in the South African context. Warm and wet months are associated with increased cattle aggregation at shared grazing areas and water points, creating conditions conducive to pathogen transmission [26]. These periods also overlap with calving and abortion events, during which Brucella organisms are shed in high concentrations into the environment, increasing opportunities for transmission [27,28]. In contrast, cooler and drier months are typically characterised by reduced animal contact intensity and lower environmental persistence of the organism, which may contribute to comparatively lower transmission risk [29,30]. However, the attenuation and change in direction of seasonal effects observed after multivariable adjustment indicate that apparent seasonal patterns are influenced not only by biological processes but also by correlated factors such as testing year, submission practices, and uneven seasonal sampling. Routine surveillance data are particularly susceptible to such effects, as testing is often reactive rather than uniformly distributed throughout the year [25,30]. Consequently, the adjusted findings do not imply a biological reduction in transmission risk during warmer months but instead highlight the challenges of disentangling true seasonal drivers from surveillance artefacts in retrospective datasets.

Marked spatial heterogeneity was observed across municipalities, with some areas consistently exhibiting higher sero-prevalence and adjusted odds of seropositivity than others. These findings indicate areas of relatively higher burden rather than confirmed transmission "hotspots," as formal spatial modelling and fine-scale geolocation data were not available. Spatial heterogeneity in bovine brucellosis has been widely documented in South Africa and elsewhere in sub-Saharan Africa and is often linked to differences in herd structure, production systems, communal grazing practices, and access to veterinary services [22,31–33]. These patterns reinforce the need for locally tailored approaches to surveillance and control rather than uniform provincial strategies.

From a programmatic perspective, the findings support the adoption of risk-based surveillance and control measures. Prioritising municipalities with consistently higher seroprevalence for enhanced testing, vaccination, and farmer engagement may improve the efficiency of resource allocation and strengthen provincial brucellosis control efforts [10,34]. In addition, although seasonal effects were attenuated after adjustment, the convergence of biological plausibility and unadjusted seasonal patterns suggests that scheduling surveillance and preventive interventions ahead of traditionally high-risk periods may still be operationally advantageous, provided such strategies are evaluated using prospectively designed data [30].

Several limitations should be considered when interpreting these findings. The analysis relied on routine laboratory submissions and on the Rose Bengal Test without systematic confirmatory testing, which may have introduced some degree of misclassification [11,35]. The retrospective nature of the dataset and the absence of herd-level information, including vaccination history, animal movement patterns, and management practices, limited causal inference and the ability to explore hierarchical or network-based transmission dynamics [24,31]. In addition, although spatial heterogeneity was evident, the lack of formal spatial modelling and detailed geospatial visualisation constrained interpretation of geographic clustering. Despite these limitations, the large sample size, multi-year coverage, and consistent analytical approach provide robust descriptive insights into the epidemiology of bovine brucellosis in Mpumalanga Province and offer a strong foundation for future prospective, spatially explicit, and One Health–oriented studies [21,22,34].

This study has several methodological strengths. First, it utilised a large provincial dataset comprising more than 67 000 cattle samples collected over four consecutive years, providing substantial statistical power and temporal coverage.

Second, the use of routine surveillance data enhances the operational relevance of the findings, as the results reflect real-world testing patterns within the provincial veterinary system. Third, the analysis incorporated both descriptive and multivariable regression approaches, allowing the independent effects of year, season, and locality to be examined while controlling for potential confounding.

However, several limitations should be considered. The study relied exclusively on routine laboratory submissions, which are not based on a probability sampling design and may therefore be influenced by outbreak-driven testing or other operational factors. The use of the Rose Bengal Test without systematic confirmatory testing may have introduced some degree of misclassification. In addition, the absence of herd-level identifiers and management data, including vaccination history, animal movements, and biosecurity practices, precluded multilevel modelling and limited causal inference. Seasonal sample sizes were also unequal, which may have affected the precision of adjusted seasonal estimates. Finally, although spatial heterogeneity was observed, the absence of fine-scale geolocation data and formal spatial modelling constrained the interpretation of geographic clustering.

Despite these limitations, the large sample size, multi-year coverage, and consistent analytical approach provide robust descriptive insights into the epidemiology of bovine brucellosis in Mpumalanga Province and offer a strong foundation for future prospective and spatially explicit studies.

## Conclusions

This study provides a province-wide description of bovine brucellosis seropositivity in Mpumalanga Province based on routine laboratory surveillance data collected between 2021 and 2024. The findings demonstrate notable temporal variability, evidence of seasonal patterning, and marked heterogeneity between local municipalities. These patterns indicate associations between seropositivity and time, season, and place, rather than causal relationships.

The transient increase in seropositivity observed during 2023, followed by a return to earlier levels, suggests episodic fluctuations in detected infection burden rather than a sustained increase over time. Seasonal differences observed in unadjusted analyses should be interpreted cautiously, as adjusted models indicate that these patterns are influenced by correlated factors such as year of testing and submission practices. Similarly, municipalities with consistently higher seroprevalence are best interpreted as areas of relatively higher burden rather than confirmed transmission hotspots, given the absence of formal spatial modelling and herd-level data.

Overall, these findings highlight the value of routine laboratory data for identifying temporal and geographic patterns that can inform risk-based surveillance and prioritisation of control activities. Future prospective studies incorporating balanced seasonal sampling, confirmatory diagnostics, herd-level metadata, and spatially explicit analyses are needed to more robustly assess drivers of transmission within a One Health framework.

## Acknowledgments

We thank the Mpumalanga Department of Agriculture, Rural Development, Land and Environmental Affairs (DARDLEA) for granting formal permission to access and analyse the dataset, and the Mpumalanga Provincial Veterinary Laboratory (MPVL) team for providing the data used in this study. We also thank the College of Agriculture and Environmental Sciences Animal Research Ethics Committee, University of South Africa (UNISA), for reviewing the study protocol and granting ethical clearance (AREC-100818–024).

## Author contributions

**Conceptualization:** Themba Titus Sigudu.

**Data curation:** Themba Titus Sigudu.

**Investigation:** Themba Titus Sigudu.

**Methodology:** Themba Titus Sigudu.

**Project administration:** Themba Titus Sigudu.

**Resources:** Themba Titus Sigudu.

**Software:** Themba Titus Sigudu.

**Supervision:** James Wabwire Oguttu.

**Validation:** Themba Titus Sigudu.

**Visualization:** Themba Titus Sigudu.

**Writing – original draft:** Themba Titus Sigudu.

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
