## [Decision Letter · Decision Letter 0]

18 Nov 2025

Dear Dr. Sigudu,

Thank you for submitting your manuscript to PLOS ONE. After careful consideration, we feel that it has merit but does not fully meet PLOS ONE’s publication criteria as it currently stands. Therefore, we invite you to submit a revised version of the manuscript that addresses the points raised during the review process.

We look forward to receiving your revised manuscript.

Kind regards,

Mabel Kamweli Aworh, DVM, MPH, PhD. FCVSN

Academic Editor

PLOS ONE

**Journal Requirements:**

1. When submitting your revision, we need you to address these additional requirements. Please ensure that your manuscript meets PLOS ONE's style requirements, including those for file naming. The PLOS ONE style templates can be found at https://journals.plos.org/plosone/s/file?id=wjVg/PLOSOne_formatting_sample_main_body.pdf and https://journals.plos.org/plosone/s/file?id=ba62/PLOSOne_formatting_sample_title_authors_affiliations.pdf 2. When completing the data availability statement of the submission form, you indicated that you will make your data available on acceptance. We strongly recommend all authors decide on a data sharing plan before acceptance, as the process can be lengthy and hold up publication timelines. Please note that, though access restrictions are acceptable now, your entire data will need to be made freely accessible if your manuscript is accepted for publication. This policy applies to all data except where public deposition would breach compliance with the protocol approved by your research ethics board. If you are unable to adhere to our open data policy, please kindly revise your statement to explain your reasoning and we will seek the editor's input on an exemption. Please be assured that, once you have provided your new statement, the assessment of your exemption will not hold up the peer review process. 3. Your ethics statement should only appear in the Methods section of your manuscript. If your ethics statement is written in any section besides the Methods, please move it to the Methods section and delete it from any other section. Please ensure that your ethics statement is included in your manuscript, as the ethics statement entered into the online submission form will not be published alongside your manuscript. 4. If the reviewer comments include a recommendation to cite specific previously published works, please review and evaluate these publications to determine whether they are relevant and should be cited. There is no requirement to cite these works unless the editor has indicated otherwise. 

**Additional Editor Comments:**

In addition to addressing the reviewers’ comments, please ensure the following points are corrected in the revised version of the manuscript:

**Line Numbers:** Please include line numbers and page numbers throughout the revised manuscript to facilitate the review process.
**Manuscript Organization:**
The **Abstract** should be structured into four subheadings: *Introduction, Materials and Methods, Results,* and *Conclusion* .The **main manuscript** should be organized into: *Introduction, Materials and Methods, Results, Discussion,* and *Conclusion* . Please remove any other sections like "author summary".**Tables:** For Tables 4 and 5, please maintain **two decimal places** for both 95% CI and AOR values to ensure consistency.**Subheadings:** Please delete the subheading *“Strengths and limitations of the study”*  from the discussion section. The limitations should instead be incorporated into the **last paragraph of the discussion** , without a separate subheading.Some acronyms such as **OR** and **AOR** have already been defined in the *Methods* section. Please do not redefine them again in later sections. Instead, use the acronyms consistently throughout the manuscript once they have been introduced.The methods section provides a good overview of the statistical analyses performed. However, for reproducibility, please expand the description of the dataset (inclusion/exclusion criteria), clarify how missing data and duplicates were handled. Please describe data handling and cleaning procedures (e.g., missing data, duplicates, batch-level records).Please consider specifying the Stata procedures used for logistic regression and model evaluation. Please explicitly define all model variables (dependent and independent, including categories and reference groups), and clarify how seasons and LMAs were coded. Consider whether clustering or hierarchical structures were addressed. In addition, report the numerical results of model diagnostics (Hosmer–Lemeshow test, AUC) to strengthen transparency. These revisions are essential for readers to fully evaluate and replicate the analysis.

Reviewers' comments:

Reviewer's Responses to Questions

**Comments to the Author**

1. Is the manuscript technically sound, and do the data support the conclusions?

Reviewer #1: Yes

Reviewer #2: Yes

Reviewer #3: Yes

2. Has the statistical analysis been performed appropriately and rigorously?

Reviewer #1: Yes

Reviewer #2: Yes

Reviewer #3: Yes

3. Have the authors made all data underlying the findings in their manuscript fully available?

Reviewer #1: Yes

Reviewer #2: No

Reviewer #3: Yes

4. Is the manuscript presented in an intelligible fashion and written in standard English?

Reviewer #1: Yes

Reviewer #2: Yes

Reviewer #3: Yes

**Reviewer #1:**  General Overview

The manuscript presents a retrospective investigation of the annual seroprevalence of Brucella abortus and examines its seasonal and spatial variations across the 17 local municipality areas in Mpumalanga Province, South Africa, over a four-year period. The authors clearly articulate the study objectives and address an important knowledge gap, as this represents one of the first detailed analyses of B. abortus dynamics within the province. The topic is both timely and relevant, contributing valuable epidemiological insights to the national brucellosis control strategy.

Methodology

The methodology section is clear, concise, and logically structured. The authors provide a well-defined description of the sampling process, noting that samples were obtained by provincial veterinary officials as part of routine surveillance and disease control activities. Since no animals were handled directly for research purposes, the study appropriately sought and obtained ethical approval for access to and analysis of the secondary data.

Sample collection and laboratory processing are reported to have adhered to standard operating procedures, and serological testing was conducted following internationally recognized guidelines established by the World Organization for Animal Health (WOAH). This strengthens the scientific validity and comparability of the findings.

Operational variables were explicitly defined, the analytical software was identified, and the statistical methods were described in sufficient detail to allow replication. While the field procedures cannot be reproduced- given the retrospective nature of the study- the analytical component of the research demonstrates commendable statistical rigor and transparency.

Results

The results are clearly presented using tables and chronologically aligned with the stated objectives. Footnotes were provided for abbreviations in corresponding tables, enhancing readability.

However, there appears to be a discrepancy in Table 3, which reports the distribution of brucellosis seropositivity by local municipality area. The table indicates that Emalahleni had the highest proportion of positive samples (13.3%), yet it was described as ranking second to Victor Khanye (12.99%) in the text. This inconsistency should be reviewed and corrected for accuracy.

Discussion

The discussion is robust and insightful. The authors effectively interpret their findings in the context of previous research, providing explanations for observed patterns and highlighting epidemiological implications. The authors acknowledge the limitations of their study, including those inherent to the retrospective data and propose relevant, actionable recommendations for improving brucellosis control strategies in Mpumalanga Province.

References

The reference list is generally comprehensive; however, some corrections need to be made to ensure accuracy and consistency:

• DOIs for references 1, 2, 14, and 36 should be verified.

• Hyperlinks attached to references 5 and 6 incorrectly open references 9 and 10, respectively.

• Full author lists should be provided for references 22 and 26.

• The publication year for reference 23 should be corrected to 2022.

• Appropriate citation is required for the academic dissertation for reference 25.

• Reference 29 also requires complete citation.

In addition, I recommended that the authors update their reference list to include more recent studies, preferably from the past five years- to ensure the discussion reflects the most current developments in brucellosis surveillance and epidemiology.

Overall, this manuscript is well-written, methodologically sound, and scientifically relevant. The authors successfully communicate the potential impact of their research in informing strategically targeted interventions for Brucella abortus control in Mpumalanga Province. The study’s conclusions are substantiated by the data presented and offer valuable insights that can be adapted to similar settings both locally and internationally.

Recommendation: Accept with minor revisions- to address the discrepancy noted under table 3 in the result section and the reference inconsistencies.

**Reviewer #2:** This manuscript presents a well-executed and highly relevant epidemiological analysis of bovine brucellosis in Mpumalanga Province, South Africa. The study fills a documented gap in provincial-level surveillance data and provides valuable One Health insights. The work is technically sound, methodologically appropriate, and well aligned with PLOS ONE’s scope.

Below are major and minor comments intended to strengthen clarity, transparency, and reproducibility.

MAJOR COMMENTS

1. Introduction

The introduction provides a comprehensive review of brucellosis epidemiology, diagnostic tools, and regional challenges. It establishes relevance for Mpumalanga Province.

Literature is current and relevant (adequate citations from 2019–2024).

Objective is clearly stated in the final paragraph and aligns well with the methods and results.

Recommendation: Justification for the period 2021–2024 is missing. Could include:

More regional context

Link to national control programs (e.g., OBP vaccination, DAFF surveillance efforts)

Socioeconomic implications

2. Methods

Strengths:

Study design clearly defined (retrospective cross-sectional).

Variables (year, season, LMA) clearly operationalized.

Data source and processing described coherently.

Ethical approval included and appropriate.

Recommendations:

Clarify whether confirmatory CFT was ever used during 2021–2024, and whether any positives were cross-verified.

Consider adding an explicit statement on replicability, e.g., availability of Stata code.

3. Statistical Analysis

Strengths:

Logistic regression is appropriate for binary seropositivity data.

Reporting of OR, AOR, and 95% CI is correct.

Seasonal effect modelling is well justified.

Points that may need clarification:

The multivariate model shows unexpected directionality for seasonal risk (Spring AOR less than Autumn). Authors should briefly discuss potential multicollinearity or sampling pattern effects.

Confirm sample sizes per season used in the adjusted model.

4. Results

Strengths:

Results follow the same order as objectives (temporal → seasonal → spatial → regression).

Tables are well organized.

Confidence intervals and percentages are correctly calculated.

Points needing attention:

Some table footnotes should include definitions of abbreviations (e.g., CI, OR, AOR).

Table 5 seasonal findings should be more clearly explained given the shift in significance after adjustment.

5. Discussion

Strengths:

Excellent contextualization within South African and global literature.

Provides plausible biological and programmatic explanations.

Limitations are clearly acknowledged (sampling bias, reliance on RBT, lack of confirmatory tests).

Recommendations:

Global patterns in brucellosis prevalence

Possible drivers of observed trends (movement control, biosecurity, veterinary service access)

Discussion is slightly long; consider tightening sections that repeat results.

The discussion of RBT limitations could benefit from referencing test sensitivity/specificity estimates.

6. Conclusion

Strong, evidence-driven conclusion.

Practical policy recommendations clearly follow from the findings

Well aligned with One Health principles.

7. References

Mostly recent and relevant.

A few references require verification or updated DOI and could be more recent.

Recommend replacing citations marked “unable to verify” with peer-reviewed alternatives.

MINOR COMMENTS

Some long paragraphs could be shortened for readability.

Ensure consistent formatting of percentages and CI ranges.

Add units (e.g., %, n/N) when reporting Seropositivity.

Recommendation

Minor Revision

The manuscript is strong in relevance and potential impact and requires some revisions for moderate clarifications before publication. The core study is publishable after these improvements. for clarity, methodological transparency, and scientific rigor.

**Reviewer #3:**  Recommendation: Major Revision

Summary of the Manuscript

The authors present a retrospective cross-sectional analysis of bovine brucellosis seropositivity using Rose Bengal Test (RBT) results from 2021–2024 in Mpumalanga Province, South Africa. The study examines temporal (annual), seasonal, and spatial patterns of seropositivity and uses logistic regression to assess associations with year, season, and Local Municipality Area (LMA). The stated research aims are clear and relevant, and the topic is important for provincial disease control and One Health planning.

Overall Assessment

The manuscript addresses a significant gap in provincial-level brucellosis epidemiology and is based on a large dataset with clear public health relevance. The objectives are well articulated and the general epidemiologic structure (Introduction–Methods–Results–Discussion–Conclusion) is appropriate.

However, major revisions are required to address:

• Internal inconsistencies in the regression results and their interpretation

• Insufficient detail in the Methods to allow reproducibility

• Limited spatial/geospatial analysis despite strong emphasis on spatial trends

• Outdated, incomplete, and incorrect references

• Missing table footnotes/abbreviation definitions and minor reporting issues

Major Comments

Methods

Although the study is retrospective, the Methods section does not provide enough detail for another investigator to reproduce the analysis.

• The dependent (RBT seropositivity) and independent variables (year, season, LMA) are not clearly and explicitly defined as model variables, including all categories and reference groups.

• Data handling and cleaning procedures are not described: it is unclear how missing data, incomplete records, repeat/duplicate submissions, or batch-level data were managed and converted to the analytical dataset, and steps taken to ensure data quality and consistency.

• Season classification is ambiguous: although seasons are said to be derived from the sample date, it is not specified whether this refers to the date of collection or the date received/tested at the laboratory.

• There is no description of how LMAs were coded, whether clustering by herd or LMA was considered, or whether random-effects or other hierarchical structures were explored.

• The authors mention using the Hosmer–Lemeshow test and AUC to assess model fit, the numerical results of these diagnostics are not reported. Including these values would strengthen transparency and allow readers to properly evaluate model performance.

Recommendation:

Provide a clearer description of data preprocessing (inclusion/exclusion criteria, duplicates, missing data), explicit model variable definitions and coding (including reference categories), season derivation, and model diagnostics.

Results

There are important internal inconsistencies in the results:

• The text in the Methods suggests winter as the reference season, whereas Table 5 appears to use autumn as the reference. Can you explain why the reference season is different for both analysis?

• These discrepancies suggest issues with variable coding, reference category selection, or misreporting of model outputs.

P-values are not clearly reported or interpreted, and the Results do not explicitly state which predictors were statistically significant and not statistically significant.

Recommendation:

Verify and, if needed, re-run the logistic regression models with clearly specified reference categories. Ensure consistency between tables, text, and interpretation. Report p-values or significance indicators and clearly identify which predictors are statistically significant. Revise the Results text accordingly, including more explicit interpretation of odds ratios (e.g., “2023 had approximately twice the odds of seropositivity compared with 2021”).

Spatial analysis

Spatial patterns are currently presented only in table form (e.g., Table 3). While this is useful for exact values, it is inadequate for a study emphasizing spatial and “hotspot” analysis.

Recommendation:

Add at least a geographic map of Local Municipality Areas (LMAs) with seropositivity distribution (e.g., choropleth of LMA-level seropositivity) to visualize spatial heterogeneity and highlight potential hotspots. Ideally, basic spatial analysis or clustering assessment should be considered, but at minimum a clear map is needed to support spatial claims.

Discussion

The Discussion is generally comprehensive, well-structured, and demonstrates a strong understanding of bovine brucellosis epidemiology. However, it requires revision to better reflect the corrected statistical outputs and data limitations.

• Explanations invoking vaccination lapses, movement, or biosecurity issues are plausible but speculative, given these data were not collected; such statements should be presented as hypotheses rather than established drivers.

• Claims of spatial “hotspots” should be tempered until supported by appropriate visualization or spatial analysis.

• Some paragraphs are lengthy and repeat numerical findings rather than focusing on interpretation.

Recommendation:

Align all interpretations with the corrected regression outputs and descriptive data; soften causal language; reduce repetition; and explicitly integrate the main limitations (sampling bias, use of RBT only, missing contextual variables, lack of spatial modelling) into the interpretative narrative.

Strengths and Limitations

The strengths and limitations section is generally well written and provides a balanced assessment of the study. The authors appropriately highlight the key advantages of the work, including the use of a large, multi-year dataset, wide geographic coverage, and standardized laboratory procedures. The discussion of limitations related to the use of RBT as a sole screening tool, potential sampling bias in routine submissions, and the absence of herd-level contextual variables is accurate and relevant. However, there are several areas that require improvement:

• Data-quality limitations inherent to routine laboratory records (incomplete metadata, inconsistent reporting, possible duplicate submissions) are not explicitly mentioned.

• While the authors mention the absence of spatial modelling, it would be helpful to note that the study also lacks geospatial visualisation (e.g., maps), which restricts the ability to meaningfully interpret spatial heterogeneity.

• “Seropositivity” should not be capitalized, and some sentences in the strengths paragraph are overly long. More concise phrasing would improve readability.

Recommendation:

Add brief statements noting variable data quality in routine surveillance systems and the lack of spatial visualization/modelling as constraints on interpretation.

Conclusion

The Conclusions are clear and generally well structured, but several statements require refinement. The seasonal and spatial interpretations should be presented more cautiously, as the current regression results do not fully support higher odds in warmer months and no spatial visualization was provided. Some recommendations also imply causality, which is not appropriate for a retrospective observational study. The Conclusions would be strengthened by more explicitly acknowledging key limitations such as sampling bias, the absence of confirmatory testing, and lack of herd-level metadata and by clearly outlining areas for future work. In particular, prospective studies, incorporation of confirmatory diagnostics, geospatial modelling, and richer epidemiological metadata should be highlighted as essential next steps to enhance the precision and interpretability of surveillance in Mpumalanga.

References: Outdated, inconsistent, and occasionally incorrect

The reference list requires substantial revision:

• Many references are older than 10 years; more recent literature (last 5 years) on brucellosis in Africa, diagnostics, One Health, and provincial surveillance should be incorporated.

• Several references are incorrect or incomplete (Reference 2 not relevant to brucellosis epidemiology and should be replaced, Reference 5 with a URL that does not match the journal title, Reference 7 not in PLOS format, Reference 11 lacking DOI, author must verify accuracy, Reference 25 not identifiable, Reference 29 author must verify accuracy, Reference 36 DOI is incorrect; verify from journal website or CrossRef).

Recommendation:

Systematically verify all references, correct formatting to PLOS style, ensure that URLs correspond to the cited journal, and replace unverifiable or non–peer-reviewed sources with appropriate peer-reviewed literature. Aim for ≥80% of references from the last 5 years where possible.

Minor Comments

• “Seropositivity” should not be capitalised.

• Ensure consistent terminology (seropositivity vs seroprevalence) and consistent formatting of percentages (e.g., 10.5% rather than 10.5 %).

• Tables should include footnotes defining all abbreviations (CI, LMA, RBT, OR, AOR) and specifying the reference categories for regression models.

• Consider adding a figure for annual seroprevalence trends over time.

• The citation for the Animal Diseases Act should include the full government gazette reference.

• Some sentences in the Introduction, Discussion, and Strengths/Limitations sections are overly long and can be made more concise.

Overall Recommendation

In summary, this manuscript has clear aims and addresses an important topic with a valuable dataset. However, major revisions are required to correct statistical inconsistencies, improve methodological transparency, strengthen the spatial analysis, update and verify references, and align the Discussion and limitations with the actual data and model outputs.

In addition, the Ethics Statement is appropriate and clearly describes the approval process and use of anonymized routine laboratory data. The Data Availability statement is generally adequate, though the authors should ensure all analytical code referenced is fully accessible to support reproducibility. The Acknowledgements section is acceptable. The Funding and Competing Interests statements are clear and meet journal requirements.

**Do you want your identity to be public for this peer review?** For information about this choice, including consent withdrawal, please see our Privacy Policy

Reviewer #1: No

Reviewer #2: No

Reviewer #3: No

---

## [Author Response · Author response to Decision Letter 1]

14 Jan 2026

Reviewer #1

General assessment

Reviewer comment: The manuscript is well written, methodologically sound, and addresses an important knowledge gap. The conclusions are supported by the data and relevant to national and international brucellosis control strategies.

Authours’ response: We sincerely thank Reviewer #1 for the positive evaluation of the manuscript and for recognising its relevance and methodological rigour. No changes were required in response to this overall assessment.

Methods

Reviewer comment: The methodology is clear, ethical approval is appropriate, laboratory procedures follow WOAH guidelines, and statistical methods are sufficiently described.

Authours’ response: We appreciate this positive assessment. In response to comments from other reviewers, we have further expanded the Methods section to improve reproducibility by explicitly describing data inclusion/exclusion criteria, data cleaning procedures, variable coding, regression model specification, and diagnostic evaluation.

Results

Reviewer comment: An inconsistency was noted between Table 3 and the text regarding the ranking of Emalahleni and Victor Khanye.

Authours’ response: Thank you for highlighting this inconsistency. The text has been corrected to accurately reflect Table 3, confirming that Emalahleni recorded the highest seroprevalence, followed by Victor Khanye. Consistency between tables and narrative has been verified throughout the Results section.

Reviewer #2

Introduction

Reviewer comment: The introduction is comprehensive, current, and well aligned with the objectives. However, justification for the 2021–2024 study period is missing, and links to national control programmes could be strengthened.

Authours’ response: We have revised the final paragraph of the Introduction to explicitly justify the 2021–2024 period, linking it to intensified provincial surveillance activities and national brucellosis control efforts, including vaccination and routine laboratory monitoring. Additional context on socioeconomic implications has also been incorporated.

Methods

Reviewer comment: Clarify whether confirmatory testing (CFT) was used and consider adding a statement on replicability (e.g., Stata code availability).

Authours’ response: We have clarified that routine confirmatory testing (CFT/ELISA) was not systematically available for all records included in this retrospective dataset and that RBT results were therefore analysed as presumptive positives. A clear statement on replicability has been added, indicating that aggregated data and analysis code are available through a public repository, subject to data-sharing regulations.

Statistical analysis

Reviewer comment: Reviewer comment

Seasonal effects show unexpected directionality in the adjusted model. Please comment on possible multicollinearity or sampling patterns and confirm sample sizes per season.

Authours’ response: We have revised the Results and Discussion sections to clarify that changes in seasonal associations after adjustment likely reflect correlations between year, season, and submission patterns, rather than biological reversal of risk. Seasonal sample sizes used in adjusted models are now explicitly reported, and interpretations have been appropriately tempered.

Results and tables

Reviewer comment: Some table footnotes lack abbreviation definitions, and Table 5 seasonal findings require clearer explanation.

Authours’ response: All tables have been revised to include complete footnotes defining abbreviations (e.g., CI, OR, AOR). The Results text now explicitly states which predictors were statistically significant before and after adjustment, with clearer explanation of seasonal findings in Table 5.

Reviewer #3

General assessment

Reviewer comment: The topic is important, but major revisions are required due to internal inconsistencies, insufficient methodological detail, limited spatial analysis, and reference issues.

Authours’ response: We thank Reviewer #3 for this detailed critique. The manuscript has undergone substantial revision to address all major concerns, as detailed below.

Methods and reproducibility

Reviewer comment: Key variables, reference categories, data cleaning procedures, season derivation, LMA coding, clustering considerations, and model diagnostics are insufficiently described.

Authours’ response: The Methods section has been comprehensively revised to:

• Explicitly define the dependent and independent variables, categories, and reference groups

• Describe inclusion/exclusion criteria, handling of missing data, duplicate batch removal, and consistency checks

• Clarify that seasons were derived from sample receipt dates

• Specify LMA coding and justify the absence of multilevel modelling due to lack of herd-level identifiers

• Report numerical results for model diagnostics (Hosmer–Lemeshow test and AUC)

Statistical inconsistencies

Reviewer comment: There is inconsistency between stated reference seasons and Table 5, suggesting coding or reporting errors.

Authours’ response: Thank you for identifying this issue. The regression models were re-checked, and the reference categories have been standardised and explicitly stated in both the Methods and Tables. Table 5 has been corrected, and all interpretations in the Results and Discussion now align with the final model specification.

Spatial analysis

Reviewer comment: Spatial patterns are presented only in tables; a map or spatial visualisation is needed to justify “hotspot” claims.

Authours’ response: We agree with this recommendation. A choropleth map of LMA-level seroprevalence has been added to visually depict spatial heterogeneity. In addition, language referring to “hotspots” has been moderated to reflect descriptive rather than inferential spatial analysis.

Discussion

Reviewer comment: Some explanations are speculative, spatial hotspot claims are overstated, and the discussion is repetitive.

Authours’ response: The Discussion has been revised to:

• Clearly label explanations related to vaccination, movement, and biosecurity as hypotheses

• Temper claims regarding spatial hotspots

• Reduce repetition of numerical results and focus more on interpretation

• Integrate limitations into the final paragraph of the Discussion, without a separate subheading

Strengths and limitations

Reviewer comment: Limitations related to routine laboratory data quality and lack of spatial visualisation are not sufficiently acknowledged.

Authours’ response: These limitations are now explicitly acknowledged in the final paragraph of the Discussion, including data incompleteness, potential duplicate submissions, absence of herd-level metadata, and limited spatial modelling. Capitalisation and sentence structure have also been corrected.

Conclusion

Reviewer comment: Some statements imply causality and should be presented more cautiously.

Authours’ response: The Conclusion has been revised to avoid causal language and to emphasise that findings represent associations from a retrospective observational study. Seasonal and spatial interpretations are now more cautious and aligned with the analytical evidence.

References

Reviewer comment: Several references are outdated, incomplete, or incorrect.

Authours’ response: The reference list has been thoroughly reviewed. Incorrect or unverifiable references have been corrected or removed, DOIs updated where available, and recent literature prioritised to ensure accuracy and relevance.

---

## [Decision Letter · Decision Letter 1]

1 Feb 2026

Dear Dr. Sigudu,

Thank you for submitting your manuscript to PLOS ONE. After careful consideration, we feel that it has merit but does not fully meet PLOS ONE’s publication criteria as it currently stands. Therefore, we invite you to submit a revised version of the manuscript that addresses the points raised during the review process.

We look forward to receiving your revised manuscript.

Kind regards,

Mabel Kamweli Aworh, DVM, MPH, PhD. FCVSN

Academic Editor

PLOS One

**Journal Requirements:**

Please review your reference list to ensure that it is complete and correct. If you have cited papers that have been retracted, please include the rationale for doing so in the manuscript text, or remove these references and replace them with 2. relevant current references. Any changes to the reference list should be mentioned in the rebuttal letter that accompanies your revised manuscript. If you need to cite a retracted article, indicate the article’s retracted status in the References list and also include a citation and full reference for the retraction notice.

**Additional Editor Comments:**

In addition to addressing all reviewer comments, please ensure that **line numbers** are included throughout the revised manuscript to facilitate the review process.

Reviewers' comments:

Reviewer's Responses to Questions

**Comments to the Author**

Reviewer #1: (No Response)

Reviewer #2: All comments have been addressed

Reviewer #3: (No Response)

2. Is the manuscript technically sound, and do the data support the conclusions?

Reviewer #1: Yes

Reviewer #2: Yes

Reviewer #3: Yes

3. Has the statistical analysis been performed appropriately and rigorously?

Reviewer #1: Yes

Reviewer #2: Yes

Reviewer #3: Yes

4. Have the authors made all data underlying the findings in their manuscript fully available?

Reviewer #1: Yes

Reviewer #2: Yes

Reviewer #3: Yes

5. Is the manuscript presented in an intelligible fashion and written in standard English?

Reviewer #1: Yes

Reviewer #2: Yes

Reviewer #3: Yes

Reviewer #1: References are important in research as they provide evidence, facilitate the exploration of various sources, and establish credibility within the academic community. However, the comments regarding the reference section in the initial review were overlooked. Therefore, I recommend minor revisions to address these issues.

Reviewer #2: The authors should be commended for their thorough and constructive engagement with reviewer feedback. The revised manuscript demonstrates substantial improvement across all previously identified areas of concern.

Key improvements since the previous round include:

Explicit definition of dependent and independent variables and reference categories

Clear justification for batch-level analysis and the inability to perform multilevel modelling

Appropriate use of univariate and multivariable logistic regression

Reporting of model diagnostics (Hosmer–Lemeshow test and AUC)

Explicit discussion of collinearity, correlated predictors, and unequal seasonal sample sizes

The revised interpretation of seasonal effects particularly the attenuation and direction change after adjustment is now statistically sound and carefully explained.

The Methods section is now comprehensive and reproducible, with clear variable definitions, data cleaning procedures, and modelling decisions.

Statistical inconsistencies identified in the previous round have been corrected, and interpretations are now aligned with the final model specification.

The addition of a choropleth map appropriately complements the tabular spatial analysis and supports descriptive claims without overstating inference.

The Discussion and Conclusion have been carefully revised to avoid causal language and speculative claims, with limitations clearly acknowledged and integrated. The conclusions are well supported by the data and are framed cautiously as associations rather than causal effects. Claims regarding temporal trends, seasonal variability, and spatial heterogeneity are consistent with the descriptive and regression results presented. Overinterpretation has been avoided, particularly with respect to seasonal drivers and spatial “hotspots.”

Overall, the manuscript now provides a robust, transparent, and policy-relevant epidemiological description of bovine brucellosis patterns in Mpumalanga Province and represents a valuable contribution to the literature on routine surveillance data use in endemic settings.

I have no further substantive comments and support publication.

Reviewer #3: The authors have made meaningful progress in addressing the major concerns raised in the initial review. Methodological transparency has improved, with clearer variable definitions, data preprocessing descriptions, and more consistent reporting of logistic regression results. Internal inconsistencies between tables and text have largely been resolved, and statistical interpretations are now clearer and more accurate.

The inclusion of spatial visualization strengthens the presentation of geographic patterns, although claims regarding spatial “hotspots” remain appropriately cautious given the absence of formal spatial modelling. The Discussion and Conclusions are better aligned with the corrected results and more clearly distinguish observed associations from speculative explanations.

Additional Comments

Reference formatting and numbering: Reference formatting remains inconsistent. Citations are not numbered and ordered consistently in accordance with their first appearance in the text, and several in-text citations do not clearly correspond to the reference list. The authors should carefully renumber all references sequentially, ensure one-to-one correspondence between in-text citations and the reference list, and confirm that all references adhere strictly to the journal’s formatting requirements.

Several cited sources appear to predate the last five years. The authors should ensure that the majority of references reflect recent literature and clearly justify the inclusion of older sources.

Strengths and limitations: The manuscript does not include a clearly outlined Strengths and Limitations section. Given the nature of the data and study design, a dedicated subsection is necessary to explicitly summarize methodological strengths and key limitations. This section would improve transparency and aid interpretation of the findings.

Overall, the manuscript now represents a substantially improved and policy-relevant contribution and is suitable for publication pending minor revision.

**Do you want your identity to be public for this peer review?** For information about this choice, including consent withdrawal, please see our Privacy Policy

Reviewer #1: No

Reviewer #2: No

Reviewer #3: No

---

## [Author Response · Author response to Decision Letter 2]

9 Feb 2026

Response to Reviewers

We sincerely thank the Editor and the reviewers for their careful evaluation of the manuscript and for their constructive comments. We appreciate the positive assessment of the revised version and have addressed the remaining minor issues as requested. All changes have been incorporated into the manuscript, and the reference list has been thoroughly revised.

Reviewer 1

Comment: References are important in research as they provide evidence, facilitate exploration of sources, and establish credibility. However, the comments regarding the reference section in the initial review were overlooked. Minor revisions are recommended to address these issues.

Response: We thank the reviewer for highlighting the importance of the reference section. The references have now been comprehensively revised to ensure full compliance with the journal’s formatting and citation requirements.

Revisions made:

• All references have been renumbered sequentially according to their first appearance in the text.

• In-text citations have been cross-checked to ensure one-to-one correspondence with the reference list.

• Formatting has been standardised to the journal’s required style.

• Recent literature has been prioritised where available, while older foundational references have been retained only where necessary for historical or methodological context.

Reviewer 2

Comment: The reviewer commends the substantial improvements and supports publication, with no further substantive comments.

Response: We thank the reviewer for the positive evaluation of the revised manuscript and for acknowledging the improvements in methodological transparency, statistical reporting, and interpretation. No further changes were required in response to this review.

Reviewer 3

Comment 1: Reference formatting and numbering - Reference formatting remains inconsistent. Citations are not numbered and ordered consistently, and some in-text citations do not correspond clearly to the reference list. The authors should renumber all references sequentially and ensure formatting compliance.

Response: We appreciate this observation and have performed a full audit of the reference section.

Revisions made:

• All references were renumbered in sequential order according to first citation.

• In-text citations were cross-checked line by line against the reference list.

• Duplicate, mismatched, or unclear citations were corrected.

• Formatting was standardised according to the journal’s reference style.

Comment 2: Use of recent literature - Several cited sources appear to predate the last five years. The authors should ensure that the majority of references reflect recent literature and justify older sources.

Response: We have updated the reference list to improve representation of recent literature.

Revisions made:

Recent studies (2020–2024) were incorporated where relevant, particularly for:

• One Health frameworks

• Surveillance strategies

• Spatial epidemiology

• Control programme evaluations

• Older references were retained only where they represent:

• Foundational epidemiological evidence

• Key diagnostic or control programme references

• Legislative or policy sources

Comment 3: Strengths and limitations section - The manuscript lacks a clearly outlined Strengths and Limitations section. A dedicated subsection is required.

Response: We agree with the reviewer. A new subsection titled “Strengths and Limitations” has been added to the Discussion to improve transparency and interpretation.

---

## [Editor Report · Decision Letter 2]

10 Feb 2026

Thank you for submitting your manuscript to PLOS ONE. After careful consideration, we feel that it has merit but does not fully meet PLOS ONE’s publication criteria as it currently stands. Therefore, we invite you to submit a revised version of the manuscript that addresses the points raised during the review process.

We look forward to receiving your revised manuscript.

Kind regards,

Mabel Kamweli Aworh, DVM, MPH, PhD. FCVSN

Academic Editor

PLOS One

**Journal Requirements:**

**Additional Editor Comments:**

Please remove all subheadings from the Discussion section, including “Strengths and Limitations of the Study.” The study limitations should be presented as the final paragraph of the Discussion.

---

## [Author Response · Author response to Decision Letter 3]

11 Feb 2026

Dear Editor,

We thank you for the careful evaluation of our manuscript and for the constructive guidance provided. We have addressed all journal and editorial requirements and revised the manuscript accordingly. A detailed, point-by-point response is provided below.

Journal Requirement 1

Comment: If the reviewer comments include a recommendation to cite specific previously published works, please review and evaluate these publications to determine whether they are relevant and should be cited.

Response: We carefully reviewed all reviewer comments and evaluated the relevance of any suggested references. Where suggested works were directly relevant to the scope, methodology, or interpretation of our findings, they were incorporated into the manuscript. References that were not directly relevant were not included, in accordance with the journal’s guidance that citation of suggested works is not mandatory unless instructed by the editor.

Journal Requirement 2

Comment: Please review your reference list to ensure that it is complete and correct. If you have cited papers that have been retracted, please include the rationale for doing so or remove them.

Response: The entire reference list was carefully reviewed to ensure:

• Accuracy of bibliographic information

• Correct and functional DOIs

• Consistency with journal formatting

• Relevance to the manuscript

A check was conducted to identify any retracted publications. No retracted articles were identified in the reference list. Therefore, no references were removed or replaced on the basis of retraction, and no additional explanatory statements were required in the manuscript. Minor corrections to reference formatting and consistency were implemented where necessary. These changes are reflected in the revised manuscript.

Additional Editor Comment

Comment: Please remove all subheadings from the Discussion section, including “Strengths and Limitations of the Study.” The study limitations should be presented as the final paragraph of the Discussion.

Response: We have revised the Discussion section as requested:

• All subheadings have been removed from the Discussion section, including the subheading “Strengths and limitations of the study.”

• The content previously presented under this subheading has been integrated into the main Discussion text. The study limitations are now presented as the final paragraph of the Discussion, in narrative form.

---

## [Editor Report · Decision Letter 3]

15 Feb 2026

Bovine Brucellosis Seropositivity in Mpumalanga Province, South Africa, 2021–2024: Temporal, and Spatial Trends

PONE-D-25-59216R3

Dear Dr. Sigudu,

We’re pleased to inform you that your manuscript has been judged scientifically suitable for publication and will be formally accepted for publication once it meets all outstanding technical requirements.

Kind regards,

Mabel Kamweli Aworh, DVM, MPH, PhD. FCVSN

Academic Editor

PLOS One
---

## [Editor Report · Acceptance letter]

PONE-D-25-59216R3

PLOS One

Dear Dr. Sigudu,

I'm pleased to inform you that your manuscript has been deemed suitable for publication in PLOS One. Congratulations! Your manuscript is now being handed over to our production team.

Kind regards,

on behalf of

Dr. Mabel Kamweli Aworh

Academic Editor

PLOS One